# P38 MAPK and Radiotherapy: Foes or Friends?

**DOI:** 10.3390/cancers15030861

**Published:** 2023-01-30

**Authors:** Natalia García-Flores, Jaime Jiménez-Suárez, Cristina Garnés-García, Diego M. Fernández-Aroca, Sebastia Sabater, Ignacio Andrés, Antonio Fernández-Aramburo, María José Ruiz-Hidalgo, Borja Belandia, Ricardo Sanchez-Prieto, Francisco J. Cimas

**Affiliations:** 1Laboratorio de Oncología Molecular, Unidad de Medicina Molecular, Centro Regional de Investigaciones Biomédicas, Unidad Asociada de Biomedicina UCLM, Unidad Asociada al CSIC, Universidad de Castilla-La Mancha, 02008 Albacete, Spain; 2Servicio de Oncología Radioterápica, Complejo Hospitalario Universitario de Albacete, 02006 Albacete, Spain; 3Servicio de Oncología Médica, Complejo Hospitalario Universitario de Albacete, 02006 Albacete, Spain; 4Departamento de Química Inorgánica, Orgánica y Bioquímica, Área de Bioquímica y Biología Molecular, Facultad de Medicina, Universidad de Castilla-La Mancha, 02008 Albacete, Spain; 5Departamento de Biología del Cáncer, Instituto de Investigaciones Biomédicas ‘Alberto Sols’ (CSIC-UAM), Unidad Asociada de Biomedicina UCLM, Unidad Asociada al CSIC, 28029 Madrid, Spain; 6Departamento de Ciencias Médicas, Facultad de Medicina, Universidad de Castilla-La Mancha, 02008 Albacete, Spain

**Keywords:** p38 MAPK, radiation, radiotherapy, MAPK14, cell cycle

## Abstract

**Simple Summary:**

p38 MAPKs comprise a family of key proteins that regulate the stress response of human cells, controlling relevant biological phenomena such as the cell cycle. As a result, their role in the cellular response to ionizing radiation and radiotherapy is crucial. In this review, we provide in-depth insights into what is known about the whole protein family’s role in response to radiation and the implications for radiotherapy.

**Abstract:**

Over the last 30 years, the study of the cellular response to ionizing radiation (IR) has increased exponentially. Among the various signaling pathways affected by IR, p38 MAPK has been shown to be activated both in vitro and in vivo, with involvement in key processes triggered by IR-mediated genotoxic insult, such as the cell cycle, apoptosis or senescence. However, we do not yet have a definitive clue about the role of p38 MAPK in terms of radioresistance/sensitivity and its potential use to improve current radiotherapy. In this review, we summarize the current knowledge on this family of MAPKs in response to IR as well as in different aspects related to radiotherapy, such as their role in the control of REDOX, fibrosis, and in the radiosensitizing effect of several compounds.

## 1. Knowing the p38 MAPK Protein Family

Mitogen-activated MAPKs are a superfamily of evolutionarily conserved serine/threonine proline target kinases (-X-Ser/Thr-Pro-X-) whose members regulate signalling cascades that convert different extracellular stimuli into a wide range of biological responses, such as survival, proliferation, differentiation or apoptosis. Among the different stimuli to which they respond are growth factors, neurotransmitters, cytokines, hormones and cellular stress [1].

In mammals, more than 14 MAPKs have been characterised and they can be classified into two groups: conventional and atypical MAPKs. While conventional MAPKs form a three-level transduction cascade and all members possess the conserved threonine-X-tyrosine (Thr-X-Tyr) motif, this hierarchical organization is lost in the atypical MAPK subfamily. The conventional MAPK group consists of extracellular signal-regulated protein kinase 1/2 (ERK 1/2), c-Jun amino (N)-terminal kinases 1–3 (JNK 1/2/3), p38 MAPKs (isoforms α, β, γ, δ), and the more recently discovered extracellular signal-regulated protein kinase 5 (ERK5). Atypical MAPKs include ERK 3/4, ERK7, ERK8 and Nemo-like kinase (NLK), whose activation cascade remains to be elucidated [2]. Regarding conventional MAPKs, the first discovered were the extracellular signal-regulated kinases (ERK) 1 and 2, which are activated mainly by growth factors, such as epidermal growth factor (EGF) or nerve growth factor (NGF) [3]. Jun N-Terminal Kinase (JNK) modules are mostly activated under conditions of cellular stress, including oxidative stress, cytokines, and UV radiation [4]. The p38 MAPKs are activated in response to different stimuli, such as oxidative stress, hypoxia, and osmotic and thermal shock, among others [5]. Finally, the most recently described module is ERK5, also known as Big MAPK1 (BMK1), whose activity increases in response to growth factors, serum, oxidative stress, and hyperosmolarity (for a review see [6]).

## 2. The p38 MAPK Pathway

The MAPK family is composed of four distinct proteins—widely labelled as isoforms—encoded by four genes: p38α (38 kDa, *MAPK14* gene, 6p21.31), p38β (41.4 kDa, *MAPK11* gene, 22q.13.33), p38ϒ (41.9 kDa, *MAPK12* gene, 22q.13.33), and p38δ (42.1 kDa, *MAPK13* gene, 6p21.31), and we refer to the four p38s collectively as p38 MAPK from now on (Figure 1). All four isoforms contain a conserved Thr-Gly-Tyr (TGY) motif that allows activation by dual phosphorylation of Thr and Tyr residues [7]. This family can be divided into two subgroups according to the percentage of homology in their amino acid sequence: p38α and p38β isoforms have 75% homology, while p38ϒ and p38δ share around 62% homology to p38α [8]. Furthermore, pharmacologically, they can be divided according to their sensitivity to pyridinyl imidazoles; p38α and p38β present a marked sensitivity to SB203580, while p38ϒ and p38δ are not affected by this compound [9]. In addition, each of the p38 MAPKs has a differential expression pattern in tissues, as well as a different substrate specificity, which may explain the diverse implications of each member in different biological processes as in the case of hematopoiesis [10]. p38α, the most abundant, and p38β are ubiquitously expressed while p38ϒ and p38δ show tissue-specific expression. p38ϒ is mainly expressed in skeletal muscle [11] and p38δ is expressed in the pancreas, kidney, small intestine, testis and lung [12]. As for its subcellular localization, p38 is ubiquitous, found both in the nucleus and in the cytoplasm under physiological conditions [13]. Following activation, accumulation of the p38α isoform has been observed in the nucleus, for example, after DNA damage [14,15]. It is also intriguing how the subcellular localization of p38α changes in response to different stimuli, as is its dependency on its own substrates, as in the case of MAPKAPK2, which is able to drive active p38α from the nucleus to the cytoplasm, thus explaining how it can phosphorylate various substrates in both compartments [14].

The best characterized member is p38α, as it was first identified in 1994 by observing tyrosine phosphorylation following endotoxin treatment, osmotic stress [16] and heat shock [17]. p38α has been identified as a key protein for the maintenance of tissue homeostasis, with involvement in different processes, ranging from differentiation to inflammatory response, and whose deregulation is implicated in several pathologies, such as neurodegenerative disorders and cancer (for a review see [18]). Despite its high homology, p38β cannot perform p38α-specific functions in embryonic development [19], and its role in acute or chronic inflammatory diseases seems to be minor [20]. Minority isoforms are less studied, although it has been recently demonstrated that their apparent secondary role is not so, with involvement in a wide range of pathologies such as cancer [21] and fungal infections [22], and whose relevance will undoubtedly increase in the coming years [23].

The p38 MAPK pathway consists of a three-kinase module in which a MAPKKK activates a MAPKK that in turn activates the MAPK by dual phosphorylation in the Thr-X-Tyr domain. There are different MAPKKKs capable of activating p38-dependent signaling, such as ASK1, TAK1, TAO and MLK3, although, in general, these MAPKKKs are promiscuous and capable of activating the JNK pathway as well [24]. In either case, canonical activation of p38 MAPK occurs through dual phosphorylation of the Thr and Tyr residues in the TGY motif by MKK3 and MKK6 kinases, leading to a conformational change that enhances protein activity and substrate recognition [25]. Although both MKK3 and MKK6 can potentially activate any of the four p38 kinases, there is evidence of some specificity between the different p38 isoforms and these MAPKKs [26] (Figure 1). However, other mechanisms of p38α activation by alternative phosphorylations have been described, such as phosphorylation at Tyr 323 by Zap70 kinase in Th1 lymphocytes following TCR receptor stimulation, allowing p38 to autophosphorylate and become self-activated [27], or activation through the ATR pathway, which is activated in response to DNA damage in cells where Cdc7 is depleted [28]. Once activated, its phosphorylation capacity will depend on its specific substrate binding sites (CD domains), characterized by the presence of hydrophobic and negatively charged residues, the subcellular location and the substrate concentration [29,30]. In fact, the sheer number of substrates may explain the plethora of effects in which p38 MAPK is involved (for a review see [31]).

Signal termination is due to a process of dephosphorylation. The involvement of the MAPK dual specific phosphatases (DUSP) has been described [32], with DUSP-1 probably being the best example [33]. Although it is not clear whether there is a specific phosphatase for each MAPK, a higher affinity for p38 MAPK has been described for some phosphatases, as in the case of PPMD1 [34], PP2A [35], and more recently DUSP-16, that is shared with JNK and seems to be implicated in cancer [36].

## 3. p38 MAPK and Cancer

The p38 MAPK pathway plays an important role in cancer initiation, progression and metastasis and has become a putative target for cancer therapy [37]. p38α is able to inhibit tumor proliferation, behaving as a tumor suppressor [38,39] through its ability to induce cell cycle arrest and senescence in response to stress stimuli [40]. In fact, it has been shown that its chemical inhibition can block tumor growth in vivo [41]. However, recent evidence demonstrates a dual role for p38α, which may behave as a tumor suppressor or as an oncogenic factor depending on the stage at which inactivation occurs [42] and with important implications for the metastatic process [43].

The role of p38α in cell cycle regulation is critical to understanding its involvement in cancer. It has been described how p38α can induce G1/S checkpoint arrest by downregulation of cyclin D1 through activation of HBP1 [44], or via direct phosphorylation of pRB, thereby increasing its affinity for the transcription factor E2F [45]. Furthermore, in contexts of DNA damage, it is known that p38α can be crucial in the activation of p53, controlling the expression of p21 and thus stopping the cell cycle in G1/S [46]. Nonetheless, p38α can also mediate G2/M arrest through targets as CDC25 or MK2 in response to DNA damage triggered by UV radiation or doxorubicin [47,48].

Its role in the tumor microenvironment is also relevant, as p38α signaling promotes tumorigenesis in lung cancer by generating a hyaluronan niche, leading to early activation of stromal fibroblasts and proliferation of cancer cells [49]. In breast cancer-associated fibroblasts, p38α promotes the activation of epithelial–mesenchymal transition (EMT) factors, such as fibronectin, N-cadherin and vimentin [50]. In addition, p38α has been proposed as a key factor in the control of other cell lineages involved in the temporal microenvironment, such as macrophages [51,52] or T cells [53].

Our knowledge of the role of the other p38 isoforms in cancer is much more limited and it is only now that we are beginning to identify their possible implications during tumorigenesis [54,55], with implications in processes such as inflammation [56] or immune system response [57]. In fact, they are already known to play a critical role in different types of tumors such as liver [21] and pancreatic cancer [58]. Those isoforms are even emerging as potential diagnostic markers in different types of tumors such as colon cancer [59], esophageal squamous cell carcinoma [60] or ovarian cancer [61].

## 4. p38 and UV Radiation

Ultraviolet (UV) radiation has been the most relevant type of radiation to which living beings have been naturally exposed and in which p38α plays a fundamental role [62]. Early studies showed that p38 MAPK was a UV-sensitive kinase [63]. p38α regulates cell cycle checkpoints in response to ultraviolet radiation by activating some of its substrates, such as CDC25 or MK2 [47,64], or more recently via its control over Claspin [65]. In addition, p38α is also involved in the repair of DNA damage caused by UV radiation. Thus, rapid activation of the p38α pathway also modifies the ubiquitination of substrates such as DDB2, facilitating the recruitment of XPC, which mediates DNA damage repair [66], or in the case of NR4A2, whose localization to sites of DNA damage is dependent on p38α signaling [67]. Although it is not known whether the activation of p38α is due to direct DNA damage or is a result of other effects generated by UV radiation, it seems to be clear that it is a determinant of cell fate [68]. In fact, in response to UV radiation, the activation of survival pathways such as AKT2 results in the attenuation of JNK and p38 activation [69], causing a decrease in UV-induced cell death and confirming the essential role of p38α in this process.

## 5. The P38 MAPK Pathway and Ionizing Radiation

Exposure to IR causes a simultaneous activation of ERK1/2, p38 MAPK, JNK [70] and also ERK5, which has recently been described as being activated in response to radiation [71,72]. This activation can be of varying intensity over time and depending on the cell type [73]. The ERK and JNK pathways appear to be in dynamic equilibrium following exposure to ionizing radiation, with the ERK pathway acting in a pro-survival role, inhibiting the JNK pathway, which acts in a pro-apoptotic role [74]. The activation of p38α by ionizing radiation was already described in the mid-1990s [75] and has been corroborated in vivo in different models [76,77]. This activation is mediated by some of its MAPKKKs, such as TAO or ASK1 [78,79]. However, it has not been specifically described which of its MAPKKs (MKK3/6) are essential for p38 MAPK activation, although it is foreseeable that both of them could be activated by IR, as occurs with most of the stimuli [26].

### 5.1. p38 MAPK and Cell Cycle in Response to IR

Activation of p38α in response to radiation results in a cellular response, leading to its translocation to the nucleus [15] and a regulation of cell cycle progression. In this sense, p38α mediates the blockade at different checkpoints such as G2/M via CDC25 [80] in which p38α activation could be mediated by TAO kinases (78). This response promotes cell cycle arrest, which, if it becomes permanent, ultimately triggers cell death [81]. p38 MAPK is also involved in the regulation of the G1/S checkpoint in response to IR through the phosphorylation of HuR, which stabilizes and triggers the cytoplasmic accumulation of p21/WAF in a p53-independent manner (Figure 2), thus mediating cell cycle arrest at this checkpoint [82]. However, it is not only p38α that has been implicated in cell cycle control in response to radiation; the MKK6-p38ϒ axis may also be involved in G2 phase control in an ATM-dependent manner [83]. It is striking that the two remaining p38 isoforms, p38β and p38δ, have not been implicated in cell cycle control in response to IR, which could indicate a certain specificity or different roles for the diverse members of the family in response to IR. 

### 5.2. p38 MAPK, ROS and Cell Fate after IR Exposure

Another effect of IR is the generation of reactive oxygen species (ROS). The role of p38α in oxidative metabolism and ROS generation has been extensively studied both within and outside the context of radiotherapy [84,85]. In fact, it is known that inhibition of p38α can affect key molecules in redox balance. It has been described how p38 MAPK is a critical mediator in radioresistance associated with COX-2 in breast cancer [86]. Furthermore, recent evidence linked P38 MAPK with DRP1, a key effector of COX2 signaling in response to IR, mediated by mitochondrial fragmentation and TFAM upregulation [86,87]. The effect of p38 MAPK on ROS signaling could explain the radiosensitizing effect of compounds like Gliotoxin [88].

In addition, the p38 MAPK pathway is a critical player in cell fate after IR exposure. In this regard, p38α-dependent signaling mediates the induction of apoptosis in response to IR in different experimental models [77,89,90,91,92] via its control of key proteins in the apoptotic process [93]. Moreover, p38 is the target of proteins that block the induction of apoptosis in response to IR, such as Mast1 or COX-2 [86,94], and it has even been proposed to also be a target of miRNAs implicated in the induction of apoptosis and senescence associated with IR [95]. However, other studies demonstrate a cytoprotective role for p38 MAPK in response to IR [95,96], or even a lack of involvement [97], suggesting that the role of p38 MAPK may be different depending on the model under study. In any case, the vast majority of studies indicate a pro-apoptotic role for p38 MAPK-mediated signaling, especially in situations of genotoxic stress [98]. Another mechanism of cellular response to IR is the induction of senescence [99]. In this context, the inhibition of the ERK and p38 MAPK pathways leads to the blocking of IR-induced β-galactosidase activity by reducing the generation of ROS [74], with p16 and p53 being two of the main effectors of p38α in the regulation of senescent phenotypes [100]. It has even been proposed that some miRNAs that mediate the induction of senescence after IR exposure do so via p38α, as in the case of miR-155 [101]. In addition, a recent study points to p38β as a mediator of the cellular response to IR through modulation of the radiation-associated senescence process [102]. In summary, there seems to be a clear consensus that the lack of signaling via p38 MAPK promotes radiation-associated senescence. Finally, in terms of cell fate, autophagy is another critical response to IR [103]. The role of p38 MAPK has not yet been characterized in autophagy triggered by IR, although in response to other genotoxic agents, such as 5-FU [104], selenite [105] or irinotecan [106], this signaling pathway has been clearly implicated. It is therefore possible that a link between p38 MAPK and radiation-triggered autophagy could be established in the future.

### 5.3. P38 MAPK and Epithelial–Mesenchymal Transition in Response to IR

The effects of p38 MAPK are not just limited to control of cell death mechanisms triggered by IR. Indeed, it has been implicated in the epithelial–mesenchymal transition (EMT) associated with IR, a key process in radioresistance [107]. In this regard, it was previously shown how p38 MAPK controls migration, a biological process associated with EMT, in response to IR in the A549 cell line [108]. It was later demonstrated how, in hepatocarcinoma cell lines, a pattern of reduced expression of E-cadherin and enhanced expressions of N-cadherin, Vimentin and Snail was found after IR exposure, which is consistent with EMT induction [109]. This effect seems to be mediated by the induction of a novel hydrogen sulfide signaling pathway that upregulates the expressions of H2S-producing proteins such as cysthionine-γ-lyase and cystathionine-β-synthase, which mediate p38α activation in response to IR. In fact, the authors showed how cysthionine-γ-lyase genetic ablation was able to prevent p38α activation and EMT in response to IR, and, more specifically, chemical inhibition of p38 MAPK was able to block IR-induced EMT in these hepatocarcinoma cell lines. In this regard, it has been proposed that MK2, one of the direct substrates of p38 MAPK, is a critical mediator in the control of IR-associated EMT exerted by p38 MAPK in head and neck cancer models [110]. Of note, it has been observed that both modulation of p38α activation and IR-induced EMT could be controlled by PARP1, as was indicated by the use of olaparib [111,112]. Therefore, all this evidence suggests that p38 MAPK is a key regulator of EMT associated with IR treatment, which renders unwanted associated effects such as the induction of metastasis or fibrosis.

### 5.4. Regulation of p38 MAPK Signaling in Response to IR

Nonetheless, it should be noted that not only the activation of p38 MAPK, but also its deactivation, is important in the response to IR. In fact, phosphatases that dephosphorylate the p38 MAPKs also appear to play an important role in the response to radiotherapy. An example of this is DUSP-1, as high expression of this phosphatase is correlated with a decrease in the effectiveness of radiotherapy by promoting DNA repair after exposure to IR through dephosphorylation of H3S10 [113]. In addition, silencing of this phosphatase leads to decreased colony formation and improved radiosensitivity in breast cancer stem cells [114]. In the case of phosphatase DUSP-16, its high expression appears to markedly decrease radiotherapy-induced apoptosis [115]. On the other hand, PP2A phosphatase is downregulated in many tumor types [116], although it has been reported that after inhibition of this phosphatase many tumors show slower growth as well as increased cell death by apoptosis and increased radiosensitivity [117]. The pharmacological approach, therefore, seems promising for regulating the phosphorylation of this pathway and thus modulating the radioresistance of different tumor types.

However, it is important to note that activation of p38 MAPK by IR is not universal. Thus, while the studies mentioned above indicate a clear activation of the pathway, others show very weak or no activation after exposure to ionizing radiation, as for example in the presence of viral genes [118]. In fact, the effect of pathway regulation could be different depending on the genetic context or the expression of certain molecules. For example, in the case of the RAS oncogenes, the expression of Ki-Ras causes a clear activation of p38 and radiosensitivity, whereas with expression of Ha-Ras, no activation of p38 is observed [119]. Another example could be TNF-α-associated radiosensitivity in lung cancer, which appears to be mediated by activation of stress response pathways, such as p38α [120]. Notably, in a murine screen for the main proteins involved in the radiation response, the p38α protein was predicted to be one of the key proteins in this process, despite the lack of changes in its expression level [121]. However, other proteomics-based screenings have shown that JNK also plays a fundamental role in this response [97]. We can therefore conclude that p38 MAPK-mediated signaling is an important effector of the cellular response to IR, but this role would depend on the cellular and genetic context in which it occurs.

## 6. Role of p38 MAPK in the Irradiation of Healthy Tissue

Not only does p38 MAPK mediate the effects on tumor tissue but also part of the effects of radiation on adjacent tissues or even further away from the tumor, which is known as the abscopal effect [122] and has been related to p38 MAPK. The exposure to low doses of radiation, such as at the tumor periphery in radiotherapy treatments, can lead to increased cell migration and invasion mediated by p38 MAPK and regulated by connexin 43 expression, which may promote metastasis formation [123]. In fact, in cells that have not been directly subjected to radiation (“bystander” cells), the activity of p38α and its substrates is increased [124]. In this regard, a recent study shows that irradiation of the thorax of mice causes a decrease in their reproductive capacity through activation of the TNFα/p38α axis, constituting another example of an abscopal effect mediated by this protein [125]. Indeed, p38α appears to play an important role in certain cellular processes related to the response to IR and not directly linked to tumor control. In endothelial cells, which are key in radiotherapy treatment for both tumor control and side effects, IR induces death by apoptosis mainly through the activation of p38α [92]. Radiation-induced cardiac damage is one of the causes of death in cancer patients receiving radiotherapy. Treatment with the cardio-protectant L-carnitine was shown to reduce radiation-induced alterations in cardiac function by decreasing myocyte apoptosis and ROS production, effects that disappeared when treated with the p38α/β inhibitor SB203580 [126]. In addition, treatment with SB203580 appears to block IL-8 overexpression after radiation exposure, clarifying the possible mechanism of action of radiation pneumonitis [127]. Furthermore, the role of p38 in myelosuppression with radiotherapy has been studied using p38 MAPK inhibitors. For example, SB203580 attenuates radiation-induced damage and senescence in hematopoietic stem cells [128] and inhibits the ROS–p38 MAPK pathway, although this is not sufficient to ameliorate long-term myelosuppression induced by total body irradiation [129]. Another study showed that adjuvant treatment with SB2003580 together with granulocyte colony-stimulating factor was able to increase hematopoiesis, mitigating the effects of total body irradiation in mice [130]. Fibrosis associated with radiation exposure in healthy tissues, as in the case of the lung [131], is one of the unresolved issues in radiotherapy. In this sense, it has been described how p38 MAPK may play an important role and its inhibition may be decisive in radiation-associated fibrosis, as has been proved in other pathological contexts such as renal fibrosis [132]. Therefore, the role of p38 MAPK in this unwanted effect of radiation should be studied in depth. In fact, there is already evidence to support the role of p38α inhibition in the control of radiation-associated fibrosis, as in the case of amifostine analogues [133]. Another type of secondary effect associated with radiotherapy is intestinal damage [134,135]. Thus, DUSP-16, a phosphatase specific to JNK and p38α [136], is implicated in this adverse effect of radiotherapy, as indicated by the use of compounds such as flagellin or rheinic acid [115,137]. Even differential activation of MAPKs, including p38 MAPK, could explain the different intestinal damage observed between sexes in experimental models [138]. In summary, p38 MAPK may play a decisive role in several aspects of radiotherapy treatment in healthy tissue, which in many cases becomes a limiting step that can lead to treatment failure.

## 7. p38 MAPK and Radiosensitivity

As mentioned above, the data available to date do not allow a definitive role to be assigned to the p38 MAPK pathway in terms of radioresistance or radiosensitivity. There are many studies in which inhibition/lack of activation correlates with a radioresistant phenotype and others in which it has no effect or even the opposite. What there does seem to be more consensus on is the potential of p38 MAPK-dependent signaling to mediate the radiosensitizing effect of different drugs. For example, p38α is fundamental to the radiosensitizing effects of 5-FU [139,140]. These observations may have certain clinical implications, as demonstrated by a retrospective analysis of 74 patients with rectal cancer that revealed a strong correlation between p38 MAPK levels and improved prognosis and response to treatment with radiotherapy combined with chemotherapy, such as 5-FU or FOLFOX, which led to increased cell death [141]. Another clear example is that of gemcitabine, although in this case it is not p38α but p38β which seems to mediate this radiosensitizing effect [142]. In addition, the p38 MAPK pathway also mediates the effect of radiosensitizers of a very different nature. One study showed that the combined treatment of arsenic trioxide with IR caused an increase in apoptosis through the generation of ROS and the activation of p38α, leading to a synergistic effect that enhanced the response of leukemia cells to IR [143]. Kale et al. have shown that the combined treatment of PU-H71, a novel Hsp90 chaperone inhibitor, with radiotherapy has a potent radiosensitizing effect that correlates with increased p38α activation [144]. In addition, the combination of berberine with radiation appears to have a synergistic effect via p38α and ROS generation [145]. Moreover, a recent study presented a synthetic alkylating agent, LQB-118, as a radiosensitizing agent in glioblastoma, which operated by reducing the total expression and phosphorylation levels of p38 [146]. Similarly, luteolin, a flavonoid, appears to increase cell death by apoptosis in combination with IR through phosphorylation and subsequent activation of the p38 MAPK pathway, among other processes [147]. Treatment with biological therapies, such as unmethylated CpG dinucleotides, has been shown to cause activation of TLR9 as well as activation of p38α, leading to increased G2/M arrest and cell death by apoptosis [148]. Another study showed how treatment of breast cancer cells with a phospholipase D inhibitor increased phosphorylation of the p38 MAPK pathway in response to radiation, and inhibition of its activation with SB203580 resulted in the loss of the increased cell death caused by the phospholipase D inhibitor in combination with radiotherapy [149]. Finally, p38α is responsible for the radiosensitivity/resistance associated with other genes; recent examples include the case of Anexin2 [81], NA methyltransferase 3A and B [150], the ubiquitin protein ligase E3 component n-recognin (UBR5) [151], or more recently, the Anterior Gradient Protein 2 homolog (AGR-2) [152]. Furthermore, p38α has been proposed to mediate the radiosensitivity associated with lncRNAs, such as TPTEP1 [153], or miRNAs, such as miRNA-153 [154]. Therefore, p38 MAPK signaling can be also a critical mediator of the radiosensitizing effects associated with different compounds, genes and other regulatory molecules, reinforcing its importance in current radiotherapy. 

## 8. Conclusions and Future Perspectives

All evidence points to an important role for the p38 MAPK signaling pathway in the response to IR, with possible clinical implications in radiotherapy. Research questions involve the role of the different isoforms, the cellular and genetic context that defines the role of p38 MAPK in the response to IR, the involvement of p38 MAPK in the different mechanisms of cell death triggered by IR, or the role in EMT associated to IR, among others. It would also be very interesting to know how p38 MAPK can control fibrosis or how p38 MAPK mediates intestinal and vascular damage in response to radiotherapy. However, the probably most relevant question encompassing all of the above is how can we improve current radiotherapy by modulating p38 MAPK expression or activity. The fact that p38 MAPK may be affecting radioresistance in irradiated tumor cells as well as its possible involvement in the effects of radiation on healthy tissue should be carefully considered for future radiotherapy treatments. To date, only one clinical trial using the highly selective p38α and p38β inhibitor ralimetinib with radiotherapy has been reported, showing good tolerability, low toxicity and encouraging results [155]. In this sense, the development of new specific inhibitors [156] or new PROTAC-based specific degraders [157] opens a new window to fully exploit the potential of p38 MAPK in radiotherapy.

## Figures and Tables

**Figure 1 cancers-15-00861-f001:**
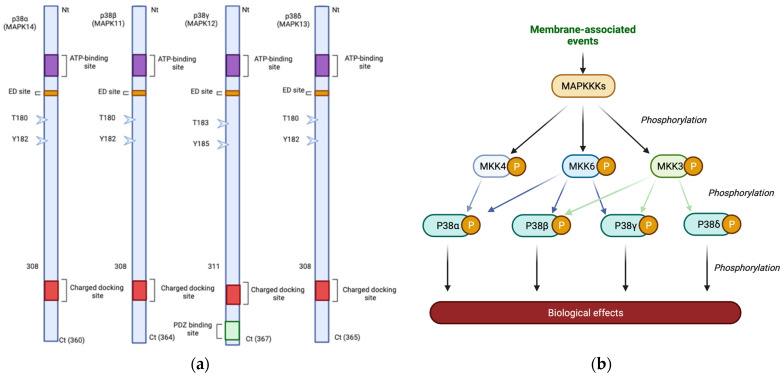
p38 MAPK diversity in structure and activation. (**a**) Diagram of the four human p38 kinases, in-dicating amino acid numbers and the different domains: The CD-charged docking domain is a negatively charged region involved in high-affinity docking interactions with positively charged substrates. The ED domain contributes to substrate docking. The ATP binding site and the phos-phorylated Thr and Tyr residues of the activation loop are also indicated. The PDZ binding site is a protein–protein interaction motif exclusive to p38γ. (**b**) Diagram of the MAP2Ks and their speci-ficity for the different p38MAPKs.

**Figure 2 cancers-15-00861-f002:**
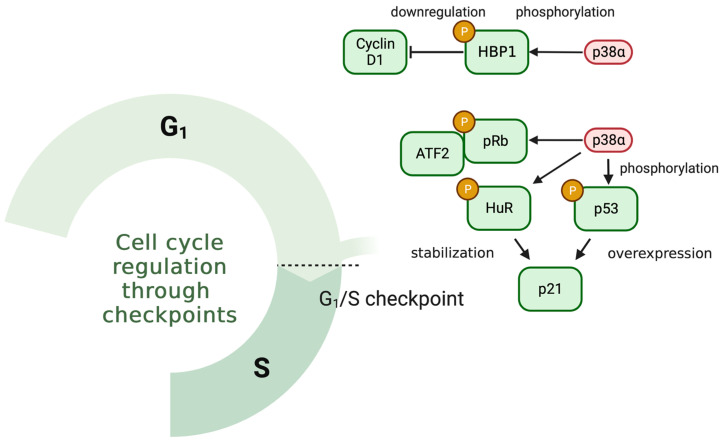
p38α controls cell cycle arrest in G1 through several mechanisms. Within its family, p38α is the main regulator of the G1/S checkpoint through different mechanisms, implicating key proteins, such as pRB, p53 and p21/WAF, in this checkpoint.

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
