# Peer review of "P38 MAPK and Radiotherapy: Foes or Friends?"

_cancers, 2023, doi:10.3390/cancers15030861_

Round 1
Reviewer 1 Report
PR’s comment on: P38 MAPK and Radiotherapy: foes or friends? By Natalia García-Flores et Al.
This reviewer hasn’t major criticisms to raise about this paper. It contributes to obviate the present paucity of publications specifically addressing the impact of radiotherapy on the p38 MAPK molecular machinery and pathways. It provides a wide and thoughtful analysis of the current knowledge on this subject, extensively covering some apparently contradictory effects regarding radio-sensitization. However, the structure of the text - although narrative - is very synthetic in reviewing the data of the literature, which makes sometimes laborious the reading. Given that the paper is not exceedingly long, a more explicative reporting could be advisable for selected subjects. In particular, this could be appreciable regarding the paragraph 5. The p38 MAPK pathway and ionizing radiation (that is, the core issue of the paper) as for cell cycle, ROS and cell fate (apoptosis, senescence, autophagy), and EMT. These subjects could be addressed in separate subparagraphs. Most important: the reference list needs to be checked for an accurate revision, due to some remarkable omissions (authors’ names, etc) in the present draft.
Author Response
Reviewer 1
This reviewer hasn’t major criticisms to raise about this paper. It contributes to obviate the present paucity of publications specifically addressing the impact of radiotherapy on the p38 MAPK molecular machinery and pathways. It provides a wide and thoughtful analysis of the current knowledge on this subject, extensively covering some apparently contradictory effects regarding radio-sensitization. However, the structure of the text - although narrative - is very synthetic in reviewing the data of the literature, which makes sometimes laborious the reading. Given that the paper is not exceedingly long, a more explicative reporting could be advisable for selected subjects. In particular, this could be appreciable regarding the paragraph 5. The p38 MAPK pathway and ionizing radiation (that is, the core issue of the paper) as for cell cycle, ROS and cell fate (apoptosis, senescence, autophagy), and EMT. These subjects could be addressed in separate subparagraphs. Most important: the reference list needs to be checked for an accurate revision, due to some remarkable omissions (authors’ names, etc) in the present draft.
We agree with the interesting suggestions of the reviewer. We have subdivided paragraph 5 and explained the subparagraphs in more detail following reviewers indications. In addition, we have carefully revised references and have corrected the missing references (4, 17, 44, 45, 48, 50).
We hope that this revised version of the manuscript will be suitable for publication in Cancers. We have re-submitted our manuscript with the new text highlited in yellow. Again, we appreciate your interest in our work as well as the comments and suggestions of both reviewers that clearly improved the quality and helped to strengthen our manuscript.
Best regards,
Francisco J. Cimas. PhD
Reviewer 2 Report
Congratulations. Very well written review of this field where knoowledge is quite extensive but still many issues remain.
Please don´t use many extra words: For example "has been shown to" of 2has been sobserved" followed by a verb, such as play and a reference. Simply "play" or the verb.
Ralimetinib shoul not be with capital letter.
Author Response
Reviewer 2
Congratulations. Very well written review of this field where knoowledge is quite extensive but still many issues remain.
Please don´t use many extra words: For example "has been shown to" of 2has been sobserved" followed by a verb, such as play and a reference. Simply "play" or the verb.
Ralimetinib shoul not be with capital letter
We appreciate the excellent comment of the reviewer. Therefore, we have carefully revised the manuscript and notably reduced the redundant expression such as "has been shown to" or “has been observed”. In addition, the capital letter in the word ralimetinib has been removed.